# Foraging Behaviour and Population Dynamics of Asian Weaver Ants: Assessing Its Potential as Biological Control Agent of the Invasive Bagworms *Metisa plana* (Lepidoptera: Psychidae) in Oil Palm Plantations

**Moïse Pierre Exélis** [1,2,*] **, Rosli Ramli** [1] **, Rabha W. Ibrahim** [3] **and Azarae Hj Idris** [1]

1 Institute of Biological Science (Biodiversity and Ecological Research Network), Faculty of Science, Universiti Malaya, Kuala Lumpur 50603, Malaysia

2 Direction de l'Enseignement Supérieur et de la Recherche Espace Etudiants, Hôtel de la Collectivité Territoriale de la Martinique (CTM), Rue Gaston Defferre, Cluny, CS 30137, Fort-de-France 97201, Martinique

3 Department of Computer Science and Mathematics, Lebanese American University, Beirut 13-5053, Lebanon

* Correspondence: exelis.pierre@gmail.com

**Abstract:** The bagworm (*Metisa plana*) is a recurrent indigenous invasive defoliator in oil palm plantations. Moderate foliar injury can cost up to 40% yield loss and more for years. The main objective of this review is to disseminate published research demonstrating the versatile services that would benefit farmers by adopting the Asian weaver ant into their pest management agenda. *Oecophylla smaragdina* is a natural indigenous enemy applied as a successful biological control agent (BCA) and strong component of integrated pest management (IPM) against important damaging pest infestations of commercial crops in the Asia-Pacific region. Farmers facing invasion could benefit by introducing *Oecophylla* ants as a treatment. The foraging behavior and population dynamics of this species are poorly documented, and hence need further evaluation. Ants of the *Oecophylla* genus, while exhibiting an intrinsic obligate arboreal pattern, demonstrate additional lengthy diurnal ground activity. The absolute territorial characteristic via continuous surveillance is significantly valuable to maintain pest balance. The exploratory scheme of major workers over large territories is derived from their inner predation instinct. The insufficient understanding of the population dynamics of this weaver ant species diverges from the knowledge of underground species. However, population density estimations of weaver ants by direct nest visual recordings are practicable and viable. The abundance assessment of individual underground ant species colonies by excavation ends with their extinction, which is not a sustainable model for *O. smaragdina*. Mathematical model estimation by simulation could not resolve this issue, adding inaccuracy to the deficiency of experimental proof. Thus, long-term monitoring of the population dynamics in real time in the field is compulsory to obtain a valid dataset. *Oecophylla* colonies, with the criteria of population stability, individual profusion, and permanent daily patrol services, are eligible as a BCA and alternative IPM treatment. The last decades have witnessed the closing of the scientific applied research gap between Asian and African species in favor of *O. longinoda* with comprehensive novel findings. By introducing *Oecophylla* ants, two main goals are reached: easing the burden of management costs for injurious insects and ending the practice of applying highly toxic pesticides that are harmful to non-target taxa, thus promoting environmental restoration.

**Keywords:** *Oecophylla* genus; population abundance; territorial foragers; quarantine defoliators; IPM

## 1. Introduction

The Asian weaver ant (*Oecophylla smaragdina*) is among the ecologically dominant insects in tropical forests, savannas [1], and agricultural landscapes [2]. It is an obligate arboreal, polydomous (multiple nests per occupied tree), absolute territorial species [3].

Few publications [4,5] have exposed the foraging and predation activities of *O. smaragdina* in oil palm plantations in Southeast Asia. The first report focused on the usage of weaver ants as a future potential biological control agent (BCA) for dominant bagworm defoliators (*Pteroma pendula*). Occupied palm trees were protected and demonstrated absence or low level foliar injury with significant higher productivity in comparison to unoccupied trees. Attack chronology patterns in relation to foraging activity were assessed in heavy infested blocks. The second report was a thesis dissertation that discussed the foraging activities of weaver ants in relation to air temperatures and relative humidity. A case study was conducted on a research national station at Teluk Intan, Perak with a preliminary study of population dynamic.

According to foraging activity, there is no major differences between Asian and African weaver ant species [6,7]. Foraging activity is a diurnal task performed exclusively by major workers, continuously patrolling outside the nests for prey along with surveillance duties [6]. Prey transportation by the foragers is performed only during the day period [8]. Infestation outbreak control largely depends on the sustainability of natural enemy populations. Thus, estimating the relative density of individuals to monitor the population dynamics of Asian weaver ants is important for effectively suppressing pests of economic importance to commercial crops [9,10].

The premise of population stability by single or assemblage species with *Oecophylla* ants achieving similar or better protection compared to specialist predators is attractive. Asian weaver ants could become a strong candidate in integrated pest management through direct application on threatened crops [11,12].

The bagworm, *Metisa plana*, an indigenous quarantine pest, is responsible for an average productivity loss of 33–40% in subsequent years of harvesting due to moderate (10–13%) foliar injury [13,14]. However, a more serious infestation can cause up to 43% yield loss over a two-year period [15]. The problems faced by smallholders and large estate plantations due to bagworm are recurrent and affect large, planted areas [16,17]. It is understood that smallholders (comprising many small plantation owners having approximately an average of 4 ha each within the same organization) are unable to properly handle outbreaks due to budget constraints. Further expansion of pest outbreaks is triggered from their small plantations to the neighboring larger cultivated area [18,19].

Plantation owners are very skeptical about using weaver ants to solve the bagworm issue owing to its pugnacious behavior towards humans [20,21]. Previous studies have showed that integrated pest management (IPM) trials [22] for treatment [23] gave conclusive successful outcomes. However, more information on weaver ants (such as mating mechanism, distinctive caste structure, population size-density, and individual behavior as a verified aggressor during foraging activity) is needed before they can be used for IPM [24] or as a BCA [25] in a large-scale management program [26,27].

This review examines other studies in order to understand weaver ant ecology. This understanding can be used to support the novel idea of bagworm control treatment by *Oecophylla* ants as a generalist predator. This review will articulate the information on *O. smaragdina:* (i) foraging behavior, (ii) population dynamics, (iii) the benefits and challengers faced by plantation owners if they adopt weaver ants to mitigate bagworm infestation, and (iv) expose recent research development towards adoption of weaver ants in agriculture and conclude some controversies, rare weaknesses, and strengths.

## 2. Research Methodology for This Review

### 2.1. Search and Assessment Inclusion Benchmark

This review was performed to collate the relevant available published academic literature. Only studies that provided information of both *Oecophylla* species were included in a first selective step. The second step of enrichment with broader sources was performed in the absence of enough supportive elements based solely on the first step criteria. Based on this review, title terms such as foraging behavior and population dynamics were the most dominant and relevant attributes to justify *Oecophylla* ants as a biological control agent. This was necessary to extract publications related only by analogy solely within ant taxonomy (http://info-now.org/ants/AntTaxonHierarchy.php, accessed on 5 October 2021) or scientific classification adhering to the Integrated Taxonomic Information System regrouping the Formicidae family. Studies written in "Bahasa Indonesian", French, Spanish, and English were included. We included studies exploring ecology, population modeling, foraging, and predation behavior. To fulfil the main objective of this review (convince farmers of the benefits), topics of the services and disservices of *Oecophylla* ants were given priority in our evaluation. Among them was the potential answer to the looming global food security crisis of including weaver ants in daily diets [28]. Finally, BCA and IPM treatments were the culminating subjects of the research findings. Tables were derived from the most relevant papers describing the associated host plant protection provided by *Oecophylla* ants from pests of economic interest: among them, classified invasive species. *O. smaragdina* was the dominant species.

### 2.2. Literature Documentation Selection

We started the literature search using the keywords "*Oecophylla* ants", "Asian weaver ants", "*Oecophylla smaragdina*", and "*Oecophylla longinoda*" in the Google search engine. The preliminary relevance of each manuscript was determined from the title based on the content of the abstract. From that initial step, if the content seemed to discuss the content of the review main topic title, we obtained its full reference, including author, year, title, and abstract, for further evaluation. We searched Google Scholar, Web of Science, frequently used databases. Because the two species of *Oecophylla* are rarely evaluated for bagworms in the oil palm plantation industry, we extended the publication date from 1960 to 2022 (articles published in the past sixty-two years), so that the review was constructed based on both older and recent literature. Considering a broader information retrieval and synthesis better demonstrated the hypothesis of *Oecophylla* ants being potent predators for the control of harmful pests. We first applied the Google general search engine to obtain different sources of papers by using keywords "*Oecophylla* foraging activity", "Asian weaver ants population", or added "dynamic", and then copy-pasted it into Google Scholar. The research was fine-tuned by adding "Scholarly articles" before each keyword. Whenever using a less specific term, such as "studies on the predatory activity of Asian weaver ants", the search turnover of 13,600 results was decreased by adding "*P. pendula*" or "*M. plana*". The decrease reached 44 and 25 potentially relevant articles, respectively, of which 15 and 4 articles abided by the intended topic title of this review. For information filtering and final selection of the manuscripts of interest, selection of the quality and eligibility of the published articles was achieved by strongly considering the following authors for most topics of study: Hölldobler & Wilson; Peng & Christian; Peng et al.; Van Mele & Cuc; Offenberg; Dejean; and Way & Khoo. By reading through pre-selected or selected articles, we found more experts doing fundamental and applied research that could significantly contribute to the value of this review as follows: Newey; Robson, Crozier, and Nielsen for the Asian species; and Nene, Vayssieres, and Rwegasira et al. for the African species. The search for keywords "*Oecophylla* foraging activity" and "*Oecophylla* population dynamic" completed the final article selection process. For foraging activity, we obtained approximately 2900 referred articles in Google Scholar, of which only 35 showed a strong relevance to the main title subject. For population dynamics, we obtained a total of 2380 results, of which only 20 were related to the manuscript title with a majority of these articles showing an orientation

for applied biological control treatment on various pests of economic interest. After initially screening the titles and reading the abstracts of an average of over 300 related articles, a total of 156 studies were identified as relevant to the title of this review: "Asian weaver ants as potential biological control agents of invasive bagworms *Metisa plana* (Lepidoptera: Psychidae): a review". For each selected article in the review, the "Related articles" option available in the Google Scholar database helped to quickly identify similar studies able to enrich the search for study inclusion in the review. For the final inclusion of identified studies, we scanned through the full-text articles to further evaluate their quality and eligibility by systematically targeting the reputable names of those researchers mentioned above that have a strong record related to the *Oecophylla* ant genus.

### 3. Foraging Behavior of Weaver Ants

Weaver ants are a well-disciplined and well-organized insect society. Its major workers caste members perform extensive foraging over a large territory to ensure the safety of the entire colony and maintain colony survival [6]. As a diurnal insect, weaver ant foragers are seen patrolling with their special task force of experienced workers to secure the whole perimeter of the colony territory [25]. Although they are strictly arboreal in nature [29,30], weaver ants have been commonly seen actively foraging on the ground [31] and moving by group of foragers [32], even when the canopies are interconnected [5].

During foraging, *Oecophylla* ants use their visual organ to detect encountered items from a distance [33] and olfactory cues to perform daily foraging duties [30]. Various authors [6,34,35] have proposed that the foraging activity of weaver ants can be summarized into five main schemes as follows: (i) the recruitment of ants into a new landscape to fill a gap in their path (i.e., obstacle crossing by bridging with more individuals). Complex chemical compounds are secreted from anus glands coupled with tactile signals. These chemicals form a chain of trails that facilitate the path of recruited nestmates using their antennae to reach the desired destination; (ii) foragers use palpable stimuli by mouth connection, antennae, or feelers, and head shaking to find resources; (iii) to explore new foraging range, fluid droplets from the rectal vesicle are laid to be detected by nestmates; (iv) to resist trespassers, an "alarm" attractant pheromone from the sternal gland is released; (v) defensive long-range recruitment comprising of odor trails, antennae, and thrilling "body jerking". All tasks related to foraging, nest guarding, and repair, along with territorial defense, are carried out by the major workers [6].

Generally, foraging and colony defense is a risky task, substantially impairing survival ability and therefore incurring high mortality rates to ants [36]. This is particularly true for *Oecophylla* ants, where major workers aggressively defend extensive territory against con-specific individuals from different colonies seen as competitors or intruders. Thus, evaluating the general activity of *Oecophylla* ants as a whole colony entity for IPM utilization is well justified. It helps in designing a better method of pest control in the field [37]. The basic main tasks at the colony level comprising the foragers' activity of major workers caste range from foraging to hunting, transporting prey items back to the nest, and surveillance [38–40].

There is still a scarcity in reports concerning the foraging activity of *O. smaragdina* or *O. longinoda* at the colony level based on 24 h monitoring scale [41]. However, another report expressed the importance of defining the appropriate daily time period to perform colony identification, transplantation, and population estimation [37]. The benefits of such manipulations will enhance integrated pest management by defining the multiple duties of weaver ants [37]. Major workers are the sole foragers outside the nest area and responsible for covering extensive grounds for hunting and predation purposes [6]. They also explore more territory to expand the colony boundaries. Figure 1A–D exposes the foraging activity of major workers on canopies, trunks, and ground in Felda oil palm plantations.

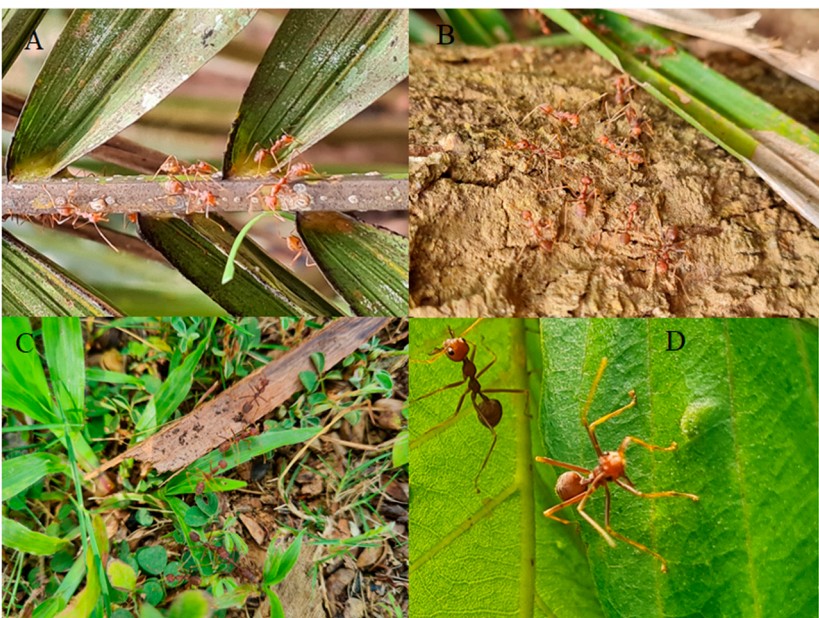

**Figure 1.** (**A**–**D**). *O. smaragdina* major workers' foraging activity: (**A**) Foragers on palm canopies frond. (**B**) Foragers on palm trunk In Felda Gunung Besout, Perak plantations. (**C**) Nomadic ground foragers around palm trees performing duties of exploring/hunting/surveillance in Felda Keratong Pahang plantations. (**D**) Foragers occupying a different plant species in oil palm plantations in Felda Keratong Pahang. Photo credit: Exélis Moïse Pierre.

## 4. Population Dynamics of Weaver Ants

After the introduction of any natural enemy, if individual abundance decreases and requires continuous artificial release upon mass-rearing to maintain its stability, this may not be economically feasible [42,43]. Therefore, the basic ecological need is the species status level, which monitors its variations in time and space [44,45]. This concept constitutes one of the main factors for a proper assessment of healthy population dynamics [46,47]. Investigations of the underlying forces (biotic and abiotic elements) responsible for those variations form the other fundamental basic components for checking and estimating PD [46]. Population dynamics are influenced by deterministic (predictable) or stochastic (unpredictable) components operating simultaneously [47]. For instance, in many insect species having short life cycles, predictable seasonal environmental parameters, such as temperature [48], rainfall interception [49], and accessible food web, influence negative or positive fluctuations in population dynamics [47]. Insects are affected by sudden variations in temperature due to their ectothermic nature [48]. The synchrony of Glanville fritillary butterfly (*Melitaea cinxia*) population dynamics during lower summer precipitation is an example of how drought affects the survival of early larvae instar, hence its metapopulation stability in the long term [49]. To successfully use weaver ants in any pest management control, it is fundamental to understand the importance of ecological factors that regulate their population dynamics. In addition, it is also compulsory to evaluate PD in the field for a long period [50]. Manipulation of the *O. smaragdina* population by introducing foreign pupae from different colonies demonstrated a successful boosting with significant worker force increase [51]. Such promotion of incipient colonies to reach growth maturity earlier than usual enables further nest translocation to targeted pest-affected crops [50].

Limited studies have been conducted directly in the field with a large agricultural monoculture over long periods of monitoring (5 to 10 years) that are backed up with empirical database records. This is because most ant colonies are subterranean. An example is the spectacular colossal intricate nest chambers (equal to the size of a house) of the attine leafcutter ant species *Acromyrmex* and *Atta* of tropical America [52]. According to ref. [53], monitoring insect taxa population dynamics by measuring their abundance and biomass

based on individual precise count is "historically" an exceptionally rare method. Nest excavation leads to colony habitat destruction [54]. Some researchers answered this hurdle by applying software simulation [55,56]. Ref. [57] gave caution on the adequacy of the ability of such models to predict and explain the overall characteristics of the collective behavior of ants by having scarce quantitative validation and insufficient experimental evidence.

Fortunately, the population dynamics of *O. smaragdina* can be estimated using the direct nest counting method (all individual castes, brood ants, and eggs) (Figure 2A,B). This method is feasible for planters and agricultural officers without the need to consider nest volume and other nest characteristics because none of the parameters are correlated to individuals' distribution in the nest [25]. Nest distribution uniformity within the same habitat or plantation for mature colonies is documented with an average occupancy comprising a range (per tree, per colony) [5,6]. Verifying that the distribution of *O. smaragdina* is not correlated to nest internal and external variables (i.e., volume), this method is acceptable [10].

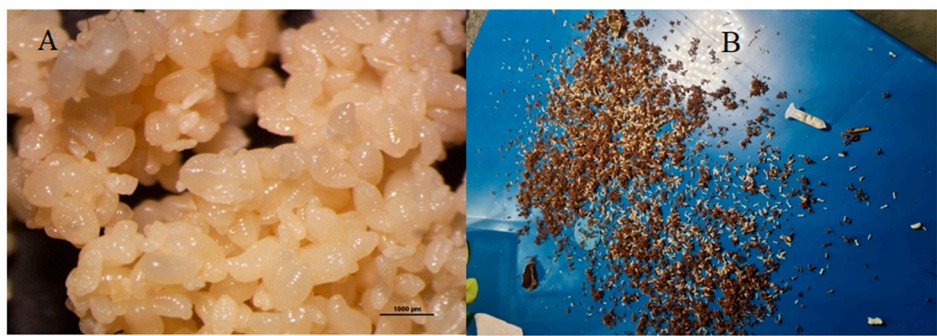

**Figure 2.** (**A,B**). Numerous *O. smaragdina* eggs from clusters extracted from a captured nest in Felda Gunung Besout, Perak oil palm plantations and examined by using a stereomicroscope Nikon SMZ800N (**A**). Brood ants, major, and minor workers exposed for direct counting of all individual castes (**B**). Photo Credit (Moïse Pierre Exélis).

As a potential BCA candidate, *O. smaragdina* is viable for practical reasons, such as abundance of individual predators versus that of defoliators [7]. Their surface occupancy is sufficient with fairly large individual numbers, enabling physical counting without needing to destroy the colony for estimation assessment. *O. smaragdina* is never subterranean, but some nests can be found on the ground under heaps of debris or piles of vegetation [31]. In addition, in Peninsular Malaysia, this method produced satisfactory results without complications [5]. This result in oil palm plantations was similar to previous reports on the abundance of individuals per colony for the *Oecophylla* genus [10,58].

In an earlier study, the population size of an *Oecophylla* colony was estimated to be approximately half a million major workers, with more than ¼ million or more brood ants, without providing data on the total number of minor workers [59]. Similar reports from other studies [60–63] have confirmed the existing range of mature colonies with an average population of several millions of workers. The population abundance and its dynamic may vary with the adopted colony's habitat, such as tropical primary or secondary rain forests, large monoculture, selected preferred variety of fruits trees, including medicinal plants, as well as rural or urban zones [9,58,64].

The widely recorded geographical distribution of *O. smaragdina* in Asia and Oceania [65,66] gives the species some reliable edge as a potential BCA in large agricultural landscapes. To sum up this concept of density interrelated functions for maintaining BCA stability [67,68], the regulation of the population size is dependent on the persistence of positive fluctuations recurrent over generations [69,70], but the lack of data asserting whether any resilient metapopulation is bound by a variable mechanism adds to the ambiguity of this ecological fundamental concept with contemporary challenges [71]. The interdependence between abiotic and biotic factors with coexisting species based on the natural principles of competition is the determinant factor [72–74].

## 5. Benefits and Challenges

*O. smaragdina* is an ecologically dominant and aggressive ant species [75]. Asian weaver ants are reputed to be excessively pugnacious generalist predators that prey on a wide range of insects [76]. Their prey comprise eight orders with twenty-six families, for a total of more than one hundred different pests [12]. Being by nature highly predaceous ants, *O. smaragdina* exhibit extensive exploratory behavior [32], with major workers having long, slender, and serrated mandibles exhibiting elongated distal teeth, perfectly adapted to their hunting inner instinct [77]. Records show that in China, *Oecophylla* nests have been introduced in citrus orchards to control pests since 300 A.D [78]. A study [79] in the Solomon Islands showed the smooth dispersal of *O. smaragdina* in coconut trees by having ants naturally infect new plantations, thus establishing new unwavering colonies.

Research has focused mainly on citrus and cashew nut crops [80], but some reports provided solutions for a variety of pests in mango orchards [22]. An integrated pest management model using Asian weaver ants in Australia to control major pests such as leafhoppers, *Idioscopus nitidulus*, red-banded thrips, *Selenothrips rubrocinctus*, the mango tip borer, *Penicillaria jocosatrix*, the fruit spotting bug, *Amblypelta lutescens lutescens*, the kernel weevil, *Sternochetus mangiferae*, the fruit fly, *Bactrocera jarvisi*, numerous leaf waves, and flower caterpillars is well documented [22]. The combined treatment of *Oecophylla* colonies with potassium soap and white oil was performed in comparison with synthetic insecticides, demonstrating a significant increase in yield without affecting pollinators [22].

In their review, ref. [12] listed only seven insect pest families susceptible to weaver ants in tropical crops. Ref. [81] reported that weaver ants significantly reduced the presence of damaging herbivores on *Rhizophora mucronata* in Thailand. Table 1 summarizes the use of Asian weaver ants as a BCA for various insect pests of economic significance affecting major crops in countries in Asia and the Pacific region. The success of using *Oecophylla* spp. as a biological control in fruit orchards (Table 1) has been well documented [81,82]. Initially, weaver ant control of numerous insect pests was associated with their diet orientation. It was suggested that the presence of the African species *O. longinoda* may have impacted the underlying mechanisms of successful pest control. Its ability to initiate the host plant to generate beneficial secondary metabolites in leaves reinforced the plants' defense against insect herbivores [83]. Furthermore, its pheromone density is recognized as a disturbing factor for oviposition by invasive *Ceratitis cosyra* and *Bactrocera invadens* (new invasive species in West-Central Africa) mango fruit flies, capable of achieving noticeable damage reduction [84]. Those pheromones were identified as having a natural fruit fly repellent effect. However, the presence of the synergistic consequence of the weaver ants using the parasitoid *Fopius arisanus* within the same ecosystem may outweigh the subsequent effective suppression on *B. invadens* (foreign invasive species on mango in Africa) [85]. Hence, this factor is important if the two natural enemies are to be adopted in combined efforts against this fruit fly species. Although weaver ants are gaining momentum as a biological control agent in Africa and Asia, there are instances where these ants are a serious hindrance for plantation workers [86]. Their ferociousness is a real nuisance during pruning and harvesting of crops [20]. A protocol helping to alleviate this problem was proposed and offered encouraging measures [80,87]. More study is needed to find a comprehensive, practical, and cheap approach to minimize the painful bites faced by maintenance staff in occupied plant canopies. The following Tables 1–7 present the results of a meta-analysis of *O. smaragdina* functions as a beneficial predator of major agricultural pests from diverse commercial crops (among them, some studies show only potential BCA treatments, see Table 3).

**Table 1.** Beneficial records of *O. smaragdina* for coconut.

| *Oecophylla* Occupied Plants (Colloquial, Scientific Name) | Control Methods—*O. smaragdina* Presence Effects | Associated Pest Species (Colloquial, Scientific Name) | Damage & Economic Yield Loss/Increase in Presence/Absence of *O. smaragdina* Treatment | Region | Key Reference(s) |
|---|---|---|---|---|---|
| Coconut *Cocos nucifera* | Satisfactory: palm base dieldrin spraying prevented *Pheidole megacephala* to induce *Oecophylla*'s population dynamic collapse. High *Oecophylla* predation on *A. cambelli*. Soursop fruit *Annona muricata* buffer zone promoted colony abundance. *Oecophylla* ants not effective on *O. arenosella*. *Monomorium floricula* and *Crematogaster* spp. outweighed *Oecophylla* egg predation. *Iridomyrmex myrmecodiae* and *Pheidole*, broke *Oecophylla* population dynamics. *Hispid* absence correlated with presence of *Oecophylla*. | *Amblypelta cocophaga* | Premature bug nut fall, sucks sap of young coconut | Solomon Islands | [89,90] |
| | | *Axiagastus cambelli* *Brontispa longissima* *Promocotheca* spp. | *A. cambelli* causes dry, thin, long nut production (no milk) | NBPNG * | [91–95] |
| | | *Opisina arenosella* | Similar | Sri Lanka | [96] |
| | | Coconut bug, *A. cocophaga* Hispine beetle *B. longissima.* Palm leaf beetle *P. papuana* & *P. opacicollis* | Young leaf feeder with seedlings and mature palm damage. | Solomon Islands | [94,97] |
| | | | Destruction of leaflet distal parts by feeding [88]: 2 years recurrent yield loss before full recovery. | Papua New Guinea | [98] |

* New Britain Papua New Guinea.

**Table 2.** *O. smaragdina* benefits for agarwood, lychee, and cocoa.

| Agarwood Gaharu *Aquilaria* spp. *Gyrinops* spp. | Control: 2–4 *Oecophylla* ants per prey | *Heortia vitessoides* | Excessive defoliation | Indonesia | [99] |
|---|---|---|---|---|---|
| Lychee *Litchi chinensis* | 1 nest managed to prevent foliar injurious insects and pentatomid bug | Lychee stink bug, *Tessaratoma papillosa.* | Fruit: premature fall, external feeding, discoloration. Inflorescence: external feeding, fall of shedding. Stems: external feeding, necrosis. | China | [78] |
| Cocoa, *Theobroma cacao* | *Oecophylla* abundant population provided complete protection *P. megacephala* control; *Oecophylla* ants effective protection. | *Helopeltis theobromae* *Amblypelta theobromae* *Peudodoniella laensisPantorhytes* spp. | Mosquito bug nymphs, adults infest cherelles, pods, young shoots. *A. theobromae*: high yield loss. Weevil borer larvae: sapwood of trunks, branches digging 1–3 cm deep burrows causing bark canker water mold disease *Phytophthora palmivora* and termites. | Malaysia PNG Solomon Islands Malaysia | [20,21,101] [102] [93,95,103] [104] [105–107] |
| | *Oecophylla* population increased by shrimps, palm sugar pellet feeding: 7.44% and 13.38% less damage, respectively. | *Pantorhytes biplagiatus* *Conopomorpha cramerella* Sn. | Severe pod damage by 21.54%, clumped beans. 64% or more yield loss with significant Average Damage Severity Index (ADSI) of 3.5 (dry season) [100] | Indonesia | |
| *T. cacao* | Highly abundant colonies reaching level 5: healthy fruit. Serious fruit damage in control plots. *O. smaragdina* absence/presence > 36–50% pod lesions, respectively. No choice feeding demonstrated highly aggressive predation. | Cocoa pod borer Conopomorph (Acrocerops) cramerella Snellen Cacao mired bug (CMB) *Helopeltis bakeri* | Similar damage as others reports Severe lesions on pods | Indonesia Philippine | [108] [109] |

**Table 3.** *O. smaragdina* benefits for citrus, *Manilkara zapota*.

| | | | | | |
|---|---|---|---|---|---|
| Citrus<br>*Citrus* spp.<br>*C. sisensis* | Buffer conservation zone of associated plants * for weaver ant abundance. | *Tessaratoma papillosa* and other Heteroptera, | *R. humeralis* sucks the juice from fruit, leaves, and branches. | China | [78,112,113] |
| *C. reticulata* | Effective with pomelo trees **. IPM by *Oecophylla* replaced WHO classified | *Rhynchocoris humeralis* | *R. serratus* principally seed-feeding and sap | Philippine | [114] |
| *C. maxima*<br>*C. sinensis* | extremely hazardous insecticides, i.e., methyl parathion. Reduced by | *R. serratus*<br>*Phyllocnistis citrella*<br>*Toxoptera aurantii—T. citricida* | feeder<br>*P. citrella* eat leaf tissues | Vietnam | [115] |
| *C. limon* | 50% pesticide use dependency & 60% vector disease. *Diaphorina citri* reduction but ineffective on mealy bugs. *** | *Diaphorina citriPanonychus citri* | *T. aurantii- citricida* responsible for *Citrus tristeza* closterovirus (CTV) a phloem virus | Vietnam<br>Thailand | [23] |
| Sapodilla-Naseberry @ *M. zapota* | IPM: Effective protection on mixed pomelo/orange equal yield, lower cost than chemical treatments. | *Phyllocoptruta oleivora*<br>*Bactrocera* spp.<br>*Eudocima salaminia* | *P. oleivora* infests mature branches, green twigs, leaves, and fruit skins causing | China | [111,116] |
| Calamansi- Limau kasturi,<br>*Citrofortunella microcarpa* | Leafhopper *Idioscopus clypealis* (honeydew) tending = Negativeproductivity in Thailand mangoorchards [23]. | *Ophiusa coronate*<br>*Hypomeces squamosus*. Asiatic citrus psyllid (ACP), *Diaphorina citri*. | heavy yield/quality losses.<br>*D. citri* cause greening disease<br>*E. salaminia* green fruit-piercing moth | Indonesia | [117] |
| | Suppressed a wide number of pest species.<br>*O. smaragdina* colonies partially present during the year (5 months). | NA [a]<br><br>25 records | *O. coronate* fruit-piercing moth<br><br>High density of curculionid beetle | India | [118] |
| | Potential BCA | | *H. squamosus* extensively grazing on young shoots, immature trees, larger trees (no ants or pesticides), See [111] | Malaysia | [110] |
| | Potential BCA | | Citrus vein phloem degeneration (CVPD) @ huánglóngbìng. 20% yield increase per year | | |
| | Predation on large number of insect pests [110] | | NA [a]<br><br><br>NA [a] | | |

* *Eucalyptus tereticorni, Ceiba pentandra, Mangifera indica, Spondias dulcis, Annona glabra, Premna integrifolia* ** Larger and thicker leaves provide better protection from cold temperatures for nest survival during winter periods. *** Mutualistic relation for honey dew energetic source. [a] Not available.

**Table 4.** *O. smaragdina* benefits for mango.

| | | | | | |
|---|---|---|---|---|---|
| Eucalyptus *Eucalyptus* spp. | Vegetation clearing around trees in combination with the introduction of *Oecophylla* colonies | *Acria cocophaga* | Adult, nymph causing heavy shoot-tip necrosis | Solomon Islands | [119] |
| Mango *Mangifera indica* | Potassium soap (1.5%)/white oil (2%) spray reduced scale/mealy bug damage on fruit [26]. Reduction of formic acid damage on fruit by separation of *Oecophylla*. Abundance of weaver ants + soft chemicals < 1% caused much lower downgraded mango than untreated orchards [120]. Control fail: honeydew producer leafhopper *I. clypealis* & *Oecophylla* association = 125% less profit compared to chemical treatments & no fruit setting [23]. No difference in presence/absence of *Oecophylla* on mealy bug *Drosicha mangiferae* & scales *Aulacaspis tubercularis* occurrence [23] | *Cryptorrhynchus gravis* *Idioscopus nitidulus* *Sternochetus mangiferae* *Campylomma austrina* *Selenothrips rubrocinctus* [a] *Bactrocera jarvisi* [23] *I. clypealis* [23] Mango leaf webber larvae *Orthaga euadrusalis* | Hoppers Nymphs & adult suck sap: panicles, tender shoots = withering and dropping of florets, lower photosynthesis.  Thrips weaken inflorescence, causing serious bronzing of the fruit surface due to the presence of air in emptied cell cavities; 20% 1st fruit class increase & 80% profit/tree per year [22].  Fruit flies maggots cause rotting | Indonesia  Australia  Thailand | [121]  [26,120]  [23] |

[a] Extra pests: Mango flower webber *Eublemma versicolor*, Mango shoot webber *Orthaga exvinacea*, Mango leaf hopper *I. nitidulus*, Red bugs *Dysdercus cingulatus*, Leaf twisting weevil *Apoderus tranquebarious* Curculionidae, Brentid beetle *Estenorhinus* spp.

**Table 5.** *O. smaragdina* benefits for Mahogany, Makassar ebony & teak wood.

| | | | | | |
|---|---|---|---|---|---|
| African mahogany *Khaya* sp. *K. ivorensis* *Swietenia macrophylla* *K. senegalensis* | Abundant colonies occupation with low cost food supplements, mix cropping, favorite host plant habitats to enhance colonies long term conservation as a maintenance buffer zone support. Comparative study with pesticides demonstrated similar or better protection, yield production Damaged trees mean average was 0–2.6% by weaver ant treatments—14.2–27.0% at Howard Springs, 28.2–48.6% at Berrimah Farm in weaver ants absence. Yearly damage trees: 4.2–32.4% by yellow loopers—0–10.4% by bush crickets | *Hypsipyla robusta*  *Amblypelta lutescens*  *Gymnoscelis* sp. *Myara yabmanna* | Shoot borer  fruit-spotting bug causing damage tree level 80–100%  25–70.4% by yellow loopers 25–100% by bush crickets | Malaysia  Australia  Australia | [58,122,123]  [25]  [37] |

**Table 5.** *Cont.*

| Makassar ebony wood *Diospyros celebica* Bakh | *Oecophylla* colonies presence maintained a low 7.69% rate of attack among 39 trees (others predators available)—important highly commercial luxury wood endemic to Sulawesi | *Arctornis submarginata Lymantria marginata* | Leaves are gnawed from the edge to the vein and midrib | Indonesia | [124] |
|---|---|---|---|---|---|
| Teak wood *Tectona grandis* | Colonies presence: control pests with an average 80–90% rate of teak trees (most important commercial wood in Indonesia). Absence of weaver ants: 30% level 1, 30% level 2, 25% level 3 and 15% level 4 foliar injury | Undetermined defoliators species *Termites Tectona grandis L.f.* | Teak stands leaves attack in early wet season | Indonesia | [125] |
| White lead tree, River tamarind *Leucaena leucocephala* | Field surveys observation records: one colony of *Oecophylla* kills an average 2000 psyllid lamtoro jumping plant lice per day. Leucanae trees widely planted offers shadows to crops and livestock feeding for animal production. Agroforestry mix systems field: Coffee—*Coffea Arabica*, Vanilla—*Vanilla planifolia*; Cocoa—*Theobroma cacao*, Oil palms—*Elaeis guineensis* Jacq. | *leucaena psyllid Heteropsylla cubana* [a] | Young leaves stems, branches, petioles gregarious feeders. USD 1.5 billion loss from five years of infestation [126,127] | Indonesia | [128] |

[a] Invasive species.

**Table 6.** Beneficial *O. smaragdina* activities in cashew nuts orchards.

| Cashew Nuts *Anacardium occidentale* | Low damage, production of high quality nuts and panicles. In absence of insecticides, ant abundance increased from 0 to 80%. First 2 or 3 years, oscillated under 80%. Colony isolation can produce 100% colonization level [129,130]. Monitoring of nest dynamic per naturally occupied cashew trees (11–13 old) [131]. Rearing colonies during 4 years with two blocks (occupied and unoccupied) by nest translocation. *O. smaragdina* nests presence throughout the year. Field, semi-field, and laboratory trials: main method used 3 weaver ant populations, i.e., 0, 5, 10 colonies per 5 plants [132]. *Oecophylla* not affecting *S. indecora* population when combined with *Helopeltis* spp. Plant protection was achieved and nymph predation occurred if cashew shoot hopper infestation occurred in absence of the tea mosquito bug [133]. | *Helopeltis pernicialis Penicillaria jocosatrix Amblypelta lutescens lutescens Anigraea ochrobasis* <br><br> *Helopeltis* spp. *H. antonii* <br><br><br><br> * *Helopeltis* spp. <br><br><br> *Sanurus indecora* | Both sap-sucking bugs 75–80% shoot, 98% flower. Positive correlation between yield with levels of ant colonization; the total variation in yield was explained by 83–90% of the results [129,130]. Tea mosquito bug damage significantly reduced with higher fruit yield. Effective protection, preying on adult and nymphs, 13.67% yield increasd [131] <br><br> Successful control of *Helopeltis* spp. population <br><br><br> Lower frequency, number of attacks on flowers of "Jambu Mete" | Australia <br><br><br><br><br><br> India <br><br><br><br><br> Sri Lanka <br><br><br><br> Indonesia | [129,130] <br><br><br><br><br><br> [131] <br><br><br><br><br> [132] <br><br><br><br> [133,134] |
|---|---|---|---|---|---|

* Main and most damaging cashew nuts pest.

**Table 7.** Medicinal and large monoculture industrial plantations.

| | | | | | |
|---|---|---|---|---|---|
| Hoop pine *Araucaria cunninghamii* | *O. smaragdina* effective larval predator | *Araucaria* looper or millionair moth *Milionia isodoxa* | Whole day larval feeding causing serious foliar injury | Papua New Guinea | [1] [135] |
| Climbing vine *Cynanchum pulchellum* | Plant used for medicinal purpose. Predation by *O. smaragdina*. | Common tiger caterpillar, *Danaus genutia genutia* | Larvae feeding on plants | Singapore | [136] |
| Oil Palm *Elaeis guineensis* Jacq. | <2% foliar injury with Level 0, 1. Higher FFB productivity/quality fruits. Bagworm infestation ≤ 15%. Weevil pollinator *Elaiedobius kamerunicus* safe. Predation trial by non-choice and choice on *S. nitens* 87.50% & 83.33% respectively | *Pteroma pendula*—*Metisa plana* [a] <br><br> *Setora nitens* *Sethosea asigna* *Thosea sinensis*, *M. plana*, *Parasa lapida* | Heavy foliar injury level 4, Less FFB productivity recorded. Severe damage/±44% yield loss/year recurrently. Foliar injury (palm canopies) | Malaysia <br><br><br><br><br> Indonesia | [4,5] <br><br> [13,14] <br><br> [137,138] |
| **Asian weaver ant** *O. smaragdina* **beneficial Synthesis** | **Ecosystem services** Productivity & yield enhancer-feces nutrient NPK [3] provider, Regulation of wide variety of pests including invasive species; Supportive of IPM helping reduce harmful pesticides dependency. Weaver ants have longer lifespan, stability factor. High pest predation rate with phytophagous regulation, fruit damage reduction, and pollination neutral effect; herbivory assists nutrient cycle | **Ecosystem disservices** Rare cases of low yield, pollinator abundance and scales insects-mealy bugs foliar damage See refs. [7,80] | **Economic Input-Societal Benefits** High income, rich nutrition, medicinal-antioxidant properties; global food crisis security component: See refs. [28,139–143] *Bacillus thuringiensis*, *Oecophylla* colony usage in IPM [2] outweigh more expensive treatments. | **Asia** | **Agricultural Systems** Variety of plantations: See refs. [7,80,144] Among all predators, *O. smaragdina* has higher potential for arboreal pest insects (CMB). |

[1] available online by 2009.   [2] Trials done on same plantations with existing colonies before Bt applications. [3] Nitrogen, Phosphorus, Potassium. [a] Invasive species.

Even though the successful application of Asian weaver ants in cocoa plantations in Malaysia was proven [101], its final adoption was recently abandoned due to the aggressive behavior of *O. smaragdina* [20,21]. This nuisance factor is impossible to avoid since man-made intervention is permanent. During the harvesting process, sometimes highly toxic synthetic chemical poisons were used to eradicate weaver ant colonies. This issue cannot be taken lightly and must be addressed [42,80]. In Africa, the use of ashes in field plantations proved to be very effective against ant bites [27]. Recently, oily repellents proved to be the most effective in Africa, hence giving promising results for eventual adoption and utilization by farmers during work [145]. Plantation staffs could carry out their duties during lowest weaver ant activity periods [37]. By avoiding the active period peaks of Asian weaver ants, pruning and harvesting were safer during early morning hours [5].

Another aspect that needs consideration is the existence of other antagonistic species observed to be a limiting factor to the normal activities of *O. smaragdina*. *Dolichoderus* spp. is suggested to impair *O. smaragdina* activities in cacao plantations in Malaysia, yet *Oecophylla* are surprisingly accepting of the presence of *Dolichoderus* with full passivity [21]. Should *O. smaragdina* to be used as a predator in oil palm plantations, this factor should be seriously considered as to not leave the ground open for any cohabitation shared between *Oecophylla*

ants and the black ant *D. thoracicus*. Repeated monitoring during surveys exposed the inability of *Oecophylla* ants to establish their colonies within occupied *D. thoracicus* areas. Another antagonistic effect of *D. thoracicus* was reported in *Citrus sinensis* and *Citrus reticulata* in which colony development of *Oecophylla* was hindered [115]. The possibility of additional existing antagonistic species is real and need further attention.

## 6. Recent Research Development towards Adoption of Weaver Ants in Agriculture

Over the recent years, a fair investment in research has been engaged in improving the management of *O. smaragdina* by incorporating innovative and effective procedures with new knowledge of the ants' ecological patterns. Such research enabled easily detecting queen nests with the purpose of future transplantation agenda [146,147]. The introduction of weaver ants in cashew nuts trees reduced the menace of the tea mosquito bug, *Helopeltis antonii* (Hemiptera: Miridae) [148]. Gravid queens of *O. smaragdina*, tested for their acceptance of foreign nest pupae, resulted in a drastic increase in the worker population within a short period of time, thus helping colonies mature faster. The proposed study conducted on incipient colonies demonstrated the viability and benefit incurred by the adoption of such new foreign broods as an early colony booster [51]. It is possible that by boosting the population dynamics from the initial stage within an undisturbed environment, those *O. smaragdina* colonies may be introduced later into crop trees [50]. The feasibility of using artificial nests to capture new queens upon nuptial flight has been demonstrated for Asian weaver ants [50]. Incipient colony development to maturity takes as long as three years to potentially produce a new queen brood. Pupae sourced from different colonies added to an incipient colony stimulates early colony growth and gains in significance if the incipient colonies are polygynous [50,51]. Larvae transplantation between different colonies is possible [82] since at this stage the nestmate recognition cues have not yet been formed [149,150]. Pupae adoption by foreign colonies of various ant species is feasible [150]. It is understood that larvae and pupae of *O. smaragdina* do not possess pheromones characteristic to determine recognition cues within each colony, which will only be acquired upon emergence of the adult stage [82]. This is beneficial for the manipulation of colony population growth, knowing that *O. smaragdina* is strictly territorial [151]. In a recent study of Asian weaver ant to promote faster early colony development, the addition of pupae to form a new colony gave promising results without hostile rejection [50]. The combination of polygyny and abundant pupae transfer achieved faster colony growth [50,51]. Comparatively, after 12 days of replacement with 60-pupae transplantation and four queens per colony, it was possible to produce much a higher brood rate than that achieved with two queens and without pupae relocation [50]. Ref. [51] demonstrated that the benefit of having a greater number of gravid queens, resulting in a drastic increase in new workers.

For long-term sustenance of a colony, the availability of food is the major dependent factor in sustainable maintenance. An experiment conducted in cashew nut orchards with *O. longinoda* demonstrated an increase in population for colonies fed additional sugar and fish protein without impeding their predatory activity [152]. Personal observation in cage culture for mass rearing demonstrated that the ants possessed exceptional survival ability when faced with severe food scarcity up to two months (continuous feeding with water, sugar, and protein was provided for the first week only). An average population size of two hundred ants was achieved. With the combination of such advantages and the beneficial factors exhibited by weaver ants, *O. smaragdina* can be a real contender in combatting pests in oil palm plantations [5]. Figure 3A,B illustrates *O. smaragdina* major workers attacking the pupae of *P. pendula* bagworms in an affected plantation [4].

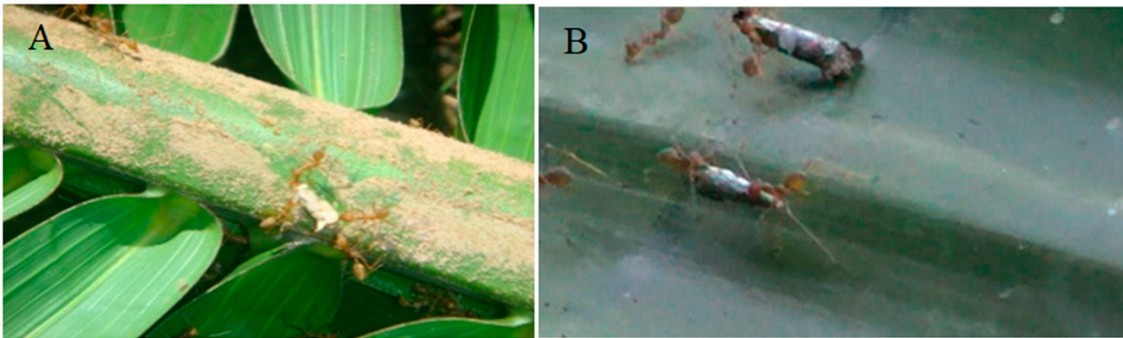

**Figure 3.** (**A**,**B**). *O. smaragdina* predation acting on *P. pendula* bagworm pupae in an oil palm plantation. (**A**) Hunting during midday peak foraging period. (**B**) Predation during late evening day period. Photo credit: Exélis Moïse Pierre.

Within the same topic, effective tested queen nurseries were recommended to save time and avoid the hassle of wild capturing ants by providing a continuous direct source of water, sugary solutions, and protein to ensure weaver ants are able to establish a new colony [153]. In the case of failure by *Oecophylla* colony treatment in the face of uncontrollable pest species, such as the mutualistic relationship between the leafhopper *Idioscopus clypealis* and weaver ants to obtain honeydew [23], it is necessary to apply alternative methods. An example of another environmentally safe application includes sex pheromone trapping and Neem (*Azadirachta indica*) application, which demonstrated compatibility with *Oecophylla* control in Ghana cocoa [154].

A recent paleontological study on *Oecophylla* fossils demonstrated an early and middle Eocene appearance from North America, with their chronological distribution related closely to ecology, behavior, and natural competition factors among global ant clusters [155]. Finally, an assessment of *Oecophylla* worker population density and dynamics is feasible using the direct nest counting method, provided that no external nest characteristics are statistically significantly correlated with the number of workers. A simplified multiple linear regression (MLR) model formula demonstrating higher accuracy performance with lower mean squared error (MSE) and root mean squared error (RMSE) has been demonstrated [156].

*Potential Controversies Weaknesses and Strengthes*

In addition to the mentioned biotic and abiotic potential factors responsible for positively or negatively influencing population dynamics, daily photoperiod cycles have never been reported to harm weaver ant colonies. *O. smaragdina* exhibiting omnivorous diet orientation [6] might invite caution about the possibility that they can prey on both herbivores and other beneficial natural enemies from surrounding crops. In fact, the Asian weaver ant demonstrated clear selective food preferences towards rich protein sources, such as live mealworms over fish, with a balanced lesser attraction for liquid-diluted honey during pilot field trials as a favourable BCA on the shoot borer, *Hypsipyla robusta* [58]. Another report exposed their predilection for chicken meat [157]. However, there is not clear reported evidence of the Asian weaver ant targeting beneficial insects in commercial crops. Few reports emphasize the risk of harming diverse pollinators in agricultural landscapes. It is opportune to revise the possibility of attack causing injuries to pollinators. Ref. [158] surveying pummelo (*C. maxima*) exposed a satisfactory and continuous attendance by diverse pollinators in the presence of *O. smaragdina* [159], which contradicted the findings of repelling pollinators due to Asian weaver ant occupancy in rambutan orchards (*Nephelium lappaceum*). This apparent setback did not disturb the fruit setting mechanism (Kazuki Tsuji, pers. com). The foraging activity of ants on *Polemonium viscosum* was suggested to promote the plant pollination process [160]. The Asian weaver ant, by targeting the less proficient pollinator species, promoted activation of the most efficient pollinators, thus providing a strategic profitable ecological service [161]. Hence, there is no evidence of weaver ants directly physically harming any pollinator. Ref. [159], by narrowing their

interpretation of "repelled pollinator", did not explore the benefits of *O. smaragdina* as a more versatile service provider [161]. The mutualistic relationship between weaver ants and trophobiont honeydew producers sheltering and feeding on plants stems damaging host plants is a rare case of disservice [12,58]. The Asian weaver ant obligate territorial stance is not derived as a preventive response to their exposed apparently weak arboreal habitat condition, but rather by natural intrinsic behaviors [6]. The major workers provide excellent protection within and beyond the colony perimeter from three dimensions (canopies, trunks, and ground), denying alien interference to their best ability by sealing all occupied sites [162]. The ground successive defensive layer mechanism, turning colony territory into a fortress of patrolling permanent major workers, is testimony to the intelligence of this species surpassing that of others [162]. The Asian weaver ant is a good natural enemy able to be introduced alone as a biological control agent against the Queensland fruit fly, *Bactrocera tryoni* by 1-octanoil emission [163], the invasive *M. plana* or incorporated as an IPM component in combination with soft chemical support [26,42]. An important slight differentiation in methodology needs to be further explained. Natural enemies are either indigenous or introduced exotic insect species (predators) already existing in various ecosystems function in the ecological balance chain as the dominant regulators of other harmful insects [164], with the second option never to be used. Such predators can be used as biological control agents by performing some artificial manipulations to help them get established and raising their population for massive propagation or for long term field nurseries, thus reaching abundant and stable levels [165]. Even though weaver ants are widely distributed, there are occurrences of poorly occupied territories in need of improvement by forming conservation buffer zones made of favorite hosts [58,78,113] or by massive translocation of their nests [25]. Ant bites are followed by the release of low quantities of formic acid, so the harm incurred by humans is not toxic. Major workers attack all stages of bagworm development, from immature to mature individuals. To conserve energy, foragers first get rid of the immobile pupae, wingless queens, and laid eggs, then all instar larvae stages by conducting a systematic prey hunt [4,5]. Few reports exposed the toxicity of pesticides to *O. smaragdina* [166]. Sometimes in large oil palm plantation monoculture, a campaign of stern elimination is conducted to suppress Asian weaver ant colonies using broad-spectrum, highly toxic, synthetic chemicals. Studies proving the effectiveness of both Asian and African weaver ants as biological control agents and IPM valuable components (combined treatment with other methods) is far beyond the infancy stage and is reaching an advanced level of achievement [7,80,144]. Hence the differentiation of the two methods is fundamental: a biological control agent is used alone for treatment control while IPM is the combination of an array of methods constitutive of biological input with soft chemicals in order to discourage the development of a harmful organism population and guarantee the lowest disruptive impact to the agro-ecosystem's health [167].

## 7. Conclusions

Weaver ants are reputable natural enemies used as a biological control agent of injurious insects to commercial crops, but a few cases have highlighted their limitation, including rupture of their population dynamics caused by competition with *D. thoracicus* along the promotion of mealy bugs and scale insects in occupied plants for mutualistic benefit. The advantages of *O. smaragdina* as a natural enemy, as a biological control agent or as a component of IPM treatment, are numerous when implemented as a side business of farming entrepreneurship. Among the various ecosystem services is the provision of NPK nutrients sourced from ant feces for absorption through leaves to be assimilated as a productivity-yield enhancer. The regulation of pests of economic interest include invasive species. The method is supportive of IPM in helping to reduce harmful pesticide dependency and weaver ants have a longer lifespan, proving a population stability factor without antagonistic ant interference. The economic input and societal benefits include side income earning, when sold for songbirds or as a nutritious diet delicacy rich in medicinal and anti-oxidants properties. Indeed, weaver ants are suggested as a potential global food

crisis security component. To implement the adoption of *O. smaragdina*, understanding its foraging activity and population dynamics is compulsory. Defining the appropriate daily time period to perform colony identification, transplantation, and population estimation will enable avoiding the nuisance of ant bites. Sustained and healthy population dynamics, corresponding to an abundance of major workers, offers more guarantee for effective pest control. It is also necessary to carry out further evaluation to close the knowledge gaps on mating behavior, colony social structure composition, and its functional activity, which are still poorly documented. In view of previous studies, conducting more field practical trials on each targeted potential pest and host plant will be valuable. The phenological differences among diverse plants need to be considered in the study's experimental design to extract more conclusive results. It is also valuable to establish buffer zone small corridors that include favorite *O. smaragdina*-occupied hosts, hence promoting the long-term conservation and population dynamics of colonies. In the last two decades, a great deal of valuable applied research towards the adoption of weaver ants has reinforced the effective biocontrol agent status of *Oecophylla* ants, including IPM applications in large or small commercial orchards. Although some setbacks have occurred due to the nuisance of ant bites in cacao plantations, the interest shown by farmers is gaining momentum. The almost cost-free application would eventually outweigh the tenacious character of ants, especially since the predator is already included by government official agencies in countries such as Australia, Africa, China, and Vietnam.

**Author Contributions:** M.P.E. proposed and conceptualized the review, analyzed the data, prepared figures and/or tables, authored or reviewed drafts of the paper. All authors contributed to consulting references, writing the paper, and approving the final draft. All authors have read and agreed to the published version of the manuscript.

**Funding:** This research was funded by the "CollectivitéTerritoriale de la Martinique-CTM" state authority in the French Island of Martinique for providing financial assistance to EMP for his doctoral study under a full scholarship European Union doctoral research grant.

**Institutional Review Board Statement:** Not applicable.

**Informed Consent Statement:** Not applicable.

**Data Availability Statement:** Not applicable.

**Conflicts of Interest:** The authors declare that they have no competing interests.

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
