# Peer review of "Foraging Behaviour and Population Dynamics of Asian Weaver Ants: Assessing Its Potential as Biological Control Agent of the Invasive Bagworms Metisa plana (Lepidoptera: Psychidae) in Oil Palm Plantations"

_sustainability, doi:10.3390/su15010780_

Round 1

Reviewer 1 Report

The bagworms Metisa plana is an indigenous invasive defoliator in oil palm plantations. The present study provided an idea of bagworms control treatment by the Oecophylla ants as generalist predator, which is interesting. The introduction could be appropriately supplemented with a little more description of the purpose and significance of the study. The conclusion could be appropriately expanded and prospective. The format of the reference section needs to be standardized and unified again. The grammatical aspect can be slightly embellished, and English should be improved further.

The manuscript needs major revision.

Some remarks are listed below:

P1,L15:“bagworm” instead of “bagworms”

P1,L17:delete “by”

P1,L18:“in Asia Pacific region”instead of“in the Asia-Pacific region”

P1,L22:add“a”between “is” and “significant”

P1,L23:“of” instead of “for”

P1,L25:“contrasts” instead of “contrast”

P1,L28:“for” instead of “from”

P1,L29:“long term” instead of “long-term”

P1,L31:“of” instead of “for”

P1,L33:“to” instead of “of”

P1,L34: “broad spectrum” instead of “broad- spectrum”

P2,L52:“According to” instead of “In term of”

P2,L58:“to” instead of “for”

P2,L58:“suppressing” instead of “suppress”

P2,L59:“of” instead of “in”

P2,L66:add “the” before “more”

P2,L68:“affect” instead of “affecting”

P2,L72:add “]”

P2,L74:“humans” instead of “human”

P2,L78:“is” instead of “are”

P2,L80:“examines” instead of “examine”

P3,L111:“to” instead of “in”

P3,L115:“pest”instead of“pests”

P3,L115:“tasks” instead of “task”

P3,L116:“foragers’ instead of “foragers”

P3,L116:“hunting” instead of “hunt”

P3,L126:“exposes” instead of “expose”

P4,L137:delete “population”

P4,L138:“factors” instead of “factor”

P4,L140:“are”instead of “is”

P4,L141:“estimating” instead of “estimate”

P4,L143:add a comma after the “instance”

P4,L143:“cycles” instead of “cycle”

P4,L144: add a comma after the “[48]”

P4,L148:“of” instead of “on”

P4,L151:“regulate” instead of “regulates”

P4,L156:“pest- affected” instead of “pest affected”

P4,L158:“are” instead of “is”

P4,L163:“individual” instead of “individuals”

P4,L163:“exceptionally” instead of “exceptional”

P4,L172:“other” instead of “others”

P4,L173:“Estimation” instead of “An estimation”

P4,L176:add “the” after the “within”

P6,L230:“being” instead of “be”

P6,L235:add “a” after the “as”

P6,L238:add“of” after the “mechanisms”

P6,L245:“using” instead of “by means of”

P6,L254:“staff” instead of “staffs”

Reviewer 2 Report

Line 144: Whether the photoperiod will affect the population dynamics?

Line 191: The literature notes are missing parentheses.

Line 229: Does Oecophylla smaragdina attack and injure pollinators?

Line 252-254: How does Oecophylla smaragdina harm humans? Is it toxic? The author should explain. 

Line 259: There are many punctuation marks before the notes.

Line 389-401: The document format is wrong with many spaces, which needs to be re-checked and modified. 

Whether Oecophylla smaragdina kills only the adults of the pest or at all stages of its development should be described in detail. 

Whether pesticides sprayed between crops are toxic to them can be explained in population dynamics.

The advantages of Oecophylla smaragdina as a natural enemy of insect pests can be described in the conclusion.

Reviewer 3 Report

In this review the authors collect the little information that exists on Foraging behavior and population dynamics of Asian weaver ants: Assessing its potential as biological control agent of the invasive bagworms Metisa plana (Lepidoptera: Psychidae) in oil palm plantations. Limited studies have been conducted directly with this ant species. Although authors do a review on the few papers that have been published however, some important issues need to be addressed by an author doing a literature review with a species like an ant.

Major comments:

1. There is a strong controversy throughout the article and that is that due to the diet of the ants, which are omnivorous, they can prey on both herbivores and other beneficial natural enemies for the surrounding crop or crops. The same authors speak of this species as “Oecophylla ants as generalist predator”. Authors should keep this fact in mind and I suggest that throughout the article the authors make an effort to emphasize that there is no information but that ecologically ants as a community are generalist predators. In addition, the authors already say that “O. smaragdina is an ecologically dominant and aggressive ant species”, if the ants are territorial, it seems that it is much more so, possibly because its nest is exposed and it is not protected underground as is the case of other species. In general, this review is very light because little information is provided, it only collects the little information that exists. I suggest that the authors add a more ecological part about the presence of this species in the crop and not limit themselves only to what is known as “possible biological control” thanks to this species. This could be done in a section before the conclusions or in them. The conclusions section seems very speculative to me.

2. On the other hand, some of the articles cited by the authors are based on very simple statistical analyzes such as correlations or regressions. I believe that with this type of analysis and studies, in addition to the small amount that exists, it is very difficult to draw a conclusion as strong as that an ant can be a good natural enemy and its population must be increased in a crop. Therefore, authors must be aware of the weaknesses of the articles they have used to carry out the review and how these articles have reached these conclusions. I suggest that the authors be more critical in this sense and be able to make a subsection on this aspect.

3. On the other hand, the authors must differentiate between “biological control agents” and “natural enemies”. This concept is basic for a researcher who writes a paper in this area of study. We speak of a biological control agent generally when they are released or their population is increased in some way in the crop. However, as described by the authors, these ants are naturally present in the crop, so we must treat them as "natural enemies".

4. Authors should also refer specifically to what would be a good “integrated pest management” and differentiate it from a “biological control agent”. Key concepts but that throughout the text seem not to be understood or specified by the authors.

I hope my comments help improve this article.

Round 2

Reviewer 1 Report

The authors have made modifications as required. Everything looks very well.

Reviewer 3 Report

I think the article is ready 
